# Is the Travel Bubble under COVID-19 a Feasible Idea or Not?

**DOI:** 10.3390/ijerph18115717

**Published:** 2021-05-26

**Authors:** Jo-Hung Yu, Hsiao-Hsien Lin, Yu-Chih Lo, Kuan-Chieh Tseng, Chin-Hsien Hsu

**Affiliations:** 1Department of Marine Leisure Management, National Kaohsiung University of Science and Technology, Kaohsiung 811532, Taiwan; henry@nkust.edu.tw; 2Department of Leisure Industry Management, National Chin-Yi University of Technology, Taichung 41170, Taiwan; chrishome12001@yahoo.com.tw (H.-H.L.); loyuchih@ncut.edu.tw (Y.-C.L.); 3MA Program in Social Enterprise and Cultural Innovation Studies, College of Humanities & Social Sciences, Providence University, Taichung 43301, Taiwan; jackt72@pu.edu.tw

**Keywords:** COVID-19, travel bubble, policy flaws, faith

## Abstract

The present study aimed to understand Taiwanese people’s willingness to participate in the travel bubble policy. A mixed research method was used to collect 560 questionnaires, and SPSS 22.0 software was used for the statistical validation and Pearson’s performance correlation analysis. Expert opinions were collected and the results were validated using multivariate analysis. Findings: People were aware of the seriousness of the virus and the preventive measures but were not afraid of the threat of infection. They looked forward to traveling to heighten their enthusiasm, relieve stress, and soothe their emotions. However, the infection and death rates have been high, there have been various routes of infection, and it has been difficult to identify the symptoms. The complex backgrounds of people coming in and out of airports, hotels and restaurants may create pressure on the participants of events. In addition, the flawed policies and high prices resulted in a loss of confidence in the policies and a wait-and-see attitude toward tourism activities. Thus, travel decisions (0.634), physical and mental health assessment (0.716), and environmental risk (−0.130) were significantly (*p* < 0.05) related to travel intentions, and different issues were affected to different degrees, while health beliefs had no significant effect (*p* > 0.05).

## 1. Introduction

Tourism is one of the major economic sources for all countries. The year 2020 has been the toughest year for the global tourism industry [1]. Since the beginning of the COVID-19 outbreak in China in December 2019, the outbreak has not yet been resolved despite the start of vaccination measures [2]. To date, country-to-country travel has been suspended around the world due to the risk of infection [3,4]. Even though governments around the world are trying to find a solution to the current situation of the tourism industry in their countries, hoping to resume tourism activities in order to revive the tourism market and business opportunities in the tourism industry [5], people’s willingness to travel is still decreasing [6] and the tourism industry is shrinking and still suffering from the impact [7,8,9]. According to the UNWTO Confidence Index, there was no sign of recovery in the tourism industry from January to April 2021, and although the birth of the vaccines has given hope [2,3,10], tourism activities are not expected to fully recover until at least after 2024 [11]. At present, it is more feasible to resume short-distance tourism trips than long-distance ones [12].

Taiwan’s is one of the closest locations to the initial COVID-19 outbreak in China. Considering the effectiveness of the outbreak control in tourist destinations, the local government in Taiwan is temporarily not recommending or opening travel to medium to high-risk areas, such as China and Vietnam, unless the destination country is assessed as a low-risk area [13]. Due to the experience in epidemic prevention, there has been no local outbreak in Taiwan and the situation is well controlled [14]. Domestic tourism activities have slightly increased [15], and the epidemic prevention measures are trusted by the general public [16]. However, there are unknown factors in the transmission of the disease, in addition to oral droplets and infectious agents, coupled with a high risk of death after infection [17,18]. At the same time, the follow-up media reports emphasize the severity of the global epidemic and reinforce fear in people’s minds [19]. Therefore, it is difficult to change people’s travel expectations in the short term [6], resulting in a significant decrease in the number of Taiwanese travelers entering and leaving the country [12,15]. The current epidemic situation in Palau is good [20], and is the same as that in Taiwan. Both governments have achieved effective results in controlling the epidemic in their territories [20,21], and both expect to recover the domestic tourism market and the overall economic situation as soon as possible [5]. The tourism bubble policy proposed by the Taiwanese government in March 2021 refers to the implementation of conditional tourism activities when the outbreak is under control and quarantine measures are mutually trusted, with the aim of shortening the quarantine period, which can also be interpreted as safe group travel [20,21]. Because the Republic of Palau (hereafter referred to as Palau) has good control of the epidemic and has diplomatic relations with Taiwan, the Taiwanese government has made it a target destination for short-haul intercountry tourism activities [20]. The travel itinerary planning includes the requirements that the participants travel in groups, that no individual itineraries are allowed, that appropriate stopping points and itineraries must be selected in advance in order to avoid crowds or specific areas and local residents, that the entire trip must be connected at designated stops, that transportation must be cleaned and disinfected daily, that hotels must have government safety and epidemic prevention certification, and that there must be a dedicated dining area for meals, a divided entry and exit route, and appropriate social distance. Participants were required to have no history of travel to or from the country within 6 months, to have not been isolated at home for the past 2 months, to have not been diagnosed with COVID-19 within 3 months, to undergo independent health management with home quarantine for 14 days after returning to the country, and to obtain a negative report for nucleic acid (PCR) testing at the airport before departure [22]. As a result, the governments of both sides have been communicating with each other in order to advocate their policies, expecting to gain public support, increase the willingness to travel, restart the development of the tourism industries in both places, and revitalize the overall economy [23]. Therefore, this study considered that the level of policy awareness can be used to estimate the public’s approval of the travel bubble policy and further estimate the willingness to participate in tourism activities.

However, even if the policy is perfect, the environmental risks of tourism still exist. The rapid scientific progress after the industrial revolution has led to the emission of large amounts of waste, resulting in an abnormal climate and more disasters, such as heatwaves, rainstorms, high temperatures, extreme cold, haze, and a proliferation of infectious diseases [24]. In addition, the unresolved COVID-19 epidemic can be transmitted by droplets and contact with infectious agents, and by other unknown routes of infection such as asymptomatic individuals, and the lethality rate is quite high [17,18,25]. Tourism activities make use of the natural landscape as well as the ecological and human–social resources to improve the physical and mental health of the participants through human planning [26]. However, the current epidemic is still unclear, and the side effects and uncertainties arising from the external risks of travel, man-made hazards, and factors such as the epidemic and climate change will increase the risk of travel and affect people’s willingness to travel [27]. Therefore, this study assumed that the awareness of environmental risks in travel can be used to estimate people’s willingness to travel.

Travel behavior originates from an idea that arises from various physical or psychological needs of an individual [28]. The idea of travel occurs when people have a need for travel. However, because of the risk of infection due to the epidemic, people’s willingness to travel has been diminished [6]. Therefore, in order to motivate individuals to travel, strong health beliefs are needed to overcome the psychological barriers. Travel health beliefs refer to an individual’s psychological state of understanding his or her own health, the assessment of his or her health alertness to the travel environment, the self-testing of healthy behaviors, and participation in healthy activities [29]. This is used by tourists to determine the risk of upcoming tourism activities and is the basis for the determination of an individual’s willingness to participate in tourism [30]. Therefore, by predicting personal behavioral intentions through health beliefs [31,32], we can analyze the degree of personal beliefs about disease control and health maintenance when traveling abroad, which can help to explore the degree of travel intentions.

Furthermore, the main purpose of tourism activities is to provide participants with a situation and behavior that improves their physical and mental health [28]. However, under the threat of an epidemic-infected environment, people’s participation in tourism behavior will constitute a risk [6]. For the public, the risk of travel is the risk of physical and mental health [33], which is contrary to the purpose of travel. Failure to protect physical and mental health will affect the willingness to participate in travel [34] and further affect travel intentions and behaviors [35]. Therefore, it is believed that conducting personal physical and mental health assessments can anticipate tourists’ willingness to participate in upcoming tourism activities and people’s confidence in travel policies.

Although travel policies are made to improve the domestic economy, the public is the main subject in generating tourism activities and promoting industry development [36]. If there is no public approval, the policy will still not be able to gain support and promote its operation [37]. Therefore, it is important to understand people’s perceptions of policies in order to obtain more in-depth answers [38,39]. Currently, there have been many studies on policy-making and environmental risk perceptions [35,36,37,38], and a number of studies on physical and mental health perceptions [37,40], as well as health beliefs [27]. Although there are studies related to decision making, environmental risks and travel intentions [41,42], there are not many studies focusing on policy-making and environmental risk cognition, health beliefs, and physical and mental health cognition related to travel intentions for international travel during an epidemic.

Furthermore, Taiwan is in line with the rest of the world in facing the threat of COVID-19 to the national economy, industry and people’s health. As a result, people’s physical and mental health are commonly affected by stress and emotions, which can lead to post-traumatic stress disorder (PTSD) [43]. Compared to other countries around the world that have begun to call for social distancing and cover-up [44], Taiwan still has some advantages in terms of actual outbreak control [10]. Therefore, the investigators believed that taking Taiwan as a case study and the local people as the target population, exploring people’s perceptions of policymaking in relation to the impact of environmental risk perception, health beliefs, and physical and mental health perceptions on travel intentions would help improve foreign travel policy planning, rebuild people’s confidence, enhance travel intentions, and restore business opportunities in the tourism industry.

## 2. Literature Discussion

### 2.1. Decision-Making Cognition

Cognition refers to the impressions received by an individual’s own perceptions, whereby a viewpoint is assigned to an environment or an object through a mental process [45]. Policy cognition is the perception of the system and the process of policy-making, as well as the perception of its promotion [10,46,47], which allows people to respond to the environment by using some basic innate behavioral patterns [45] to gain a sense of the effectiveness of policy development through the mental process in which the consciousness recognizes and understands things, such as perception, imagination, recognition, reasoning, and judgment [48,49].

Policy-making is aimed at improving the plight of local and industrial development, and is most profoundly experienced by the people whose situation is expected to improve; their feelings are usually apparent only after the policy has been implemented [10,48]. Policy cognition can be discussed in terms of the cognition of policy regulations and connotations, government planning and support measures, industry measures, personal policy recognition, and expected policy effectiveness [10,37,49]. The higher the level of policy recognition, the higher the chance of participation in the policy [10,37].

### 2.2. Environmental Risk Perception

Environmental risk refers to the process of understanding the unpredictable but potentially far-reaching potential problems that arise when people or society as a whole are exposed to environmental hazards [50], which deviate from, or even contradict, everyday knowledge, i.e., risk [51]. The degree to which an individual reacts to risk is their environmental risk perception [52].

The three elements of uncertainty, the possibility of loss, and a futuristic nature are required for a risk to be established [53]. The higher the uncertainty about the future outcome, the higher the risk [54]. This can be explored in terms of the surrounding living and transportation environment, indoor or outdoor places, dining and consumption moments, and personal feelings [53,54,55].

### 2.3. Health Beliefs

Health consciousness refers to an individual’s heightened concern for health, increased search for health information, devotion to health concerns and the importance of health status [56]. It involves self-concern similar to self-awareness, provides motivation for self-monitoring and health awareness, and induces a sense of health engagement [57]. In contrast, behavioral responses based on health awareness for individuals to accept preventive measures, which can be considered as intervention indicators [58,59], are called health beliefs [60].

Health beliefs can determine behavioral intentions related to medical products or related activities by the severity, susceptibility, barriers, and benefits of engaging in behaviors that may be health-related [60]. The level of risk can influence people’s behavior in engaging in activities [61].

### 2.4. Physical and Mental Health

Physical and mental health refers to a state of wellbeing in physical, mental and social aspects [62]. Physical and mental health can be analyzed by means of self-perception assessment [63], and can be presented in practice through scientific evidence from tests such as self-assessment [64].

The investigation of individual physical and mental health phenomena according to personal feelings can present the impact of the current environment on people [10]. Physical and mental health can be divided into three dimensions: psychological, spiritual, and attitudinal [10,63,64,65,66], which are evidenced by feelings of anxiety, competence, enthusiasm, headaches, abdominal pain, insomnia, stomach pain, an abnormal diet, and death-seeking ideation [66,67,68]. How good one feels about the current state of one’s physical and mental health affects one’s willingness to act and make judgments [10,48].

### 2.5. Travel Intentions

The intention is an individual’s tendency to anticipate, plan, or intend whether a future behavior will be performed [69], and can be used as a predictor of future behavior [70]. It can be seen as the degree to which people have the tendency to want to travel in order to fulfill their travel beliefs [71], which can be used to determine the degree to which an individual is inclined to participate in travel behavior for a particular future travel activity [72]. Hence, the focus on tourists’ travel intentions can help us to adjust the current strategies for better tourism policies for countries which are trying to revive their tourism industry [73].

Travel intention is an indicator that aims to understand the extent to which individuals travel and consider traveling [74], and to explore the degree of behavioral propensity to travel to a place [75]. It can be examined in terms of the individual’s intention to travel, the level of information learned, and the personal behavior and necessary expertise required to prepare for the travel behavior in advance [75,76].

## 3. Methods

### 3.1. Study Design, Population and Setting

The present study examines the relationship among travel decisions, environmental risk perceptions, health beliefs, physical and mental health perceptions, and the willingness of the public to participate in tourism bubble policies, so as to predict people’s perceptions of the policies and provide suggestions for the government and tourism-related industries to improve their current decisions or future measures, as shown in Figure 1.

According to the research framework description, 9 research hypotheses were proposed:

**Hypothesis** **1** **(H1).***It is assumed that people have the same cognition on tourism decision-making*.

**Hypothesis** **2** **(H2).***It is assumed that the public has the same perception of environmental risks*.

**Hypothesis** **3** **(H3).***It is assumed that people have the same perception of health beliefs*.

**Hypothesis** **4** **(H4).***It is assumed that the people’s personal physical and mental health assessment is consistent*.

**Hypothesis** **5** **(H5).***It is assumed that the people’s travel intentions are the same*.

**Hypothesis** **6** **(H6).***Tourism decision-making cognition and tourism willingness have a significant impact*.

**Hypothesis** **7** **(H7).***It is assumed that environmental risk perception and travel intentions have a significant impact*.

**Hypothesis** **8** **(H8).***It is assumed that health beliefs and travel intentions have a significant impact*.

**Hypothesis** **9** **(H9).***It is assumed that physical and mental health and travel intentions have a significant impact*.

Adequate theoretical support is needed for research studies. However, rare or novel research directions have a weak theoretical foundation. This study, therefore, adopted a complex approach, complementing the breadth of the study with quantitative research [77,78] and adding depth to the study with qualitative research [79], in order to compensate for methodological or theoretical shortcomings [80]. Using an online questionnaire platform, the researchers selected an initial sample of 1000 people who signed up for the first travel bubble event on 1 April 2021, between 20 March and 5 April 2021, through intentional sampling [81]. As it is difficult to accurately estimate the population size, this study concluded that the sample size should be at least 272 questionnaires (α = 0.05, sampling error 3%) [82]. Due to the constraints of the respondents’ willingness to be interviewed, the researcher expanded the sample by random sampling, targeting Taiwanese individuals who had experience abroad and were willing to participate in travel bubble activities. Of the 600 questionnaires obtained by the random sampling method, 560 valid questionnaires were finally analyzed after eliminating the invalid ones [83]. IBM SPSS Statistics for Windows, Version 22.0 (IBM, Armonk, NY, USA) statistical software was used to inspect the reliability and validity of the questionnaire, and then the data were analyzed by statistical validation, Pearson’s correlation method, and regression analysis. Interviews were then conducted in order to obtain insights on the analysis results. Subsequently, the information was compiled, organized and analyzed to construct the content of the report [84]. Finally, the multivariate verification analysis method was used to integrate the information of different research subjects, research theories and methods, and obtain accurate knowledge and meanings by comparing the research results from multiple perspectives and multiple points of data [85,86].

### 3.2. Measurements

The questionnaire was administered on a 5-point Likert scale, with a score of 1 being very dissatisfied and 5 being very satisfied. After the content was prepared with reference to the literature, three experts were sought to examine the content, IBM SPSS Statistics for Windows, Version 22.0. statistical software was used to determine the topic, and then a statistical validation was performed. When the Kaiser-Meyer-Olkin (KMO) was greater than 0.05 and the *p*-value in Bartlett’s test was less than 0.01 (*p* < 0.01), the scale was considered suitable for continuous factor analysis [85]. Questions with good reliability [87] with an alpha greater than 0.60 were adopted for subsequent analysis.

There were 7 questions on travel decision cognition, and the results of the statistical analysis showed that the KMO was 0.860, Bartlett’s approximate χ^2^ value was 1249.956, and the df was 21, with a significance of *p* < 0.001, which was suitable for factor analysis. The explained variance of the scale was 63.07%, and the total explained variance was 63.07%. After the factor analysis, all of them were retained. The questionnaire was named Tourism Decision Making Awareness, with 7 questions, of which the alpha coefficients were 0.903–0.915, and the alpha coefficient of the total scale was 0.921. The above analysis results showed that this questionnaire had good reliability.

There were 6 questions on environmental risk perception, and the results of the statistical analysis showed that the KMO was 0.880, and Bartlett’s approximate χ^2^ value was 1161.964 with a df of 15, which was significant at *p* < 0.001 and suitable for factor analysis. The explained variance of the scale was 67.59%, and the total explained variance was 67.59%. After the factor analysis, all of them were retained. The questionnaire was named the Environmental Risk Awareness Questionnaire, with a total of 6 questions, of which the alpha coefficients were 0.897–0.919, and the alpha coefficient of the total scale was 0.922. Based on the results of the above analysis, this questionnaire had good reliability.

There were 11 questions on health beliefs, and the results of the statistical analysis showed that the KMO was 0.735, Bartlett’s approximate χ^2^ value was 932.114, and the df was 66, with a significance of *p* < 0.001, which was suitable for factor analysis. The explained variances of the scales were 27.18% and 19.12%, and the total explained variance was 46.3%. All of them were retained after the factor analysis. The questionnaires were named Perceived Susceptibility and Seriousness (5) and Self-efficacy (6), with a total of 11 questions, of which the alpha coefficients were 0.699–0.720 and 0.725–0.754, and the alpha coefficient of the total scale was 0.760. Based on the above analysis results, this questionnaire had good reliability.

There were 9 questions on physical and mental health assessment, and the results of the statistical analysis showed that the KMO was 0.927, Bartlett’s approximate χ^2^ value was 3476.010, and the df was 36, with a significance of *p* < 0.001, which was suitable for factor analysis. The explained variances of the scales were 46.07%, 41.61% and 2.54%, and the total explained variance was 90.2%. All of them were retained after the factor analysis. The questionnaires were named Psychological Feelings (3), Mental State (3), and Attitude toward Life and Health (3), with a total of 9 questions, of which the alpha coefficients were 0.974, 0.972 and 0.973, and the overall scale alpha coefficient was 0.980. Based on the above analysis results, this questionnaire had good reliability.

There were 4 questions on travel intention, and the results of the statistical analysis showed that the KMO was 0.845 and Bartlett’s approximate χ^2^ value was 860.021, with a df of 6, and the significance was *p* < 0.001, which was suitable for factor analysis. The explained variance of the scale was 81.74%, and the total explained variance was 81.74%. After the factor analysis, all of them were retained. The questionnaire was named the Travel Intention questionnaire, with 4 questions, of which the alpha coefficients were 0.904–0.935 and the alpha coefficient of the total scale was 0.940. Based on the above analysis results, this questionnaire had good reliability. As shown in Table 1.

Nevertheless, a more deliberate and sound approach is needed to examine national public health issues. Using a hybrid research approach which collects additional information in a variety of ways for comparison or corroboration can yield more accurate and in-depth knowledge [10,37,80,81,82,83]. Therefore, after obtaining the questionnaire sample data for the analysis, the present study used video software or telephone interviewing to obtain the views of in-service tour guides or scholars with expertise in tourism and public health-related fields. The information was compiled, organized, and analyzed in a rigorous sequence in order to construct this paper [84], and the final analysis and study were conducted by multivariate validation [86,87].

### 3.3. Study Scope and Limitations

The study was conducted to investigate the water literacy of people in Taiwan. During the sampling period, the study was faced with the threat of the COVID-19 epidemic, as well as funding, manpower and material constraints. Therefore, the questionnaires on travel decision making and environmental risk perception, health beliefs, and the physical and mental health assessment were developed and distributed through an online questionnaire platform and an intentional sampling method to separate respondents, and then the random sampling method was used to expand the sampling scope. After the data were collected and analyzed, video conferencing and telephone interviews were conducted with the consent of the respondents, including tour leaders and guides, as well as experts in leisure, tourism, medical and public health, to provide insights into the analysis and results. Finally, all of the data were compiled and examined using a multivariate verification approach. The shortcomings of the study will be used as research recommendations, and we expect that follow-up researchers will conduct investigations to improve the shortcomings. As shown in Table 2.

### 3.4. Ethical Considerations

The data for this study were collected by the intentional sampling of island residents in Taiwan, combined with a random sampling method. All of the study procedures and data results were in accordance with ethical standards [88,89]. The respondents were people living in Taiwan, as well as their relatives and friends. In addition, all of the surveys and interviews were conducted after the respondents understood the purpose of the study and agreed to be interviewed. The interview results and related data were also provided on site for analysis with their permission.

## 4. Analysis and Discussion

The present study analyzed Taiwanese people’s willingness to participate in the travel bubble policy in terms of decision-making and environmental risk perception, health beliefs, and physical and mental health. In total, 560 samples were collected for analysis. The subjects’ perceptions were tested using basic statistical tests with SPSS 26.0 software (M = mean, Rank = mean rank). The data were analyzed by Pearson’s correlation test and a linear regression test to verify whether there is an interaction between travel decisions, environmental risk, physical and mental health, and travel intention perception. Six experts and scholars were interviewed to provide their insights based on the analysis. All of the data were then compiled for categorization, generalization and comparison [83], and were explored in a multivariate approach [84,85].

### 4.1. Background

The sample background was analyzed by statistical analysis. The majority of the respondents were male (61.1%) and the lowest number of respondents were female (38.9%). The age of the respondents was 51–60 (21.7), followed by 21–30 (20.5%), 41–50 (19.7%), 31–40 (19.3%), under 20 (12.7%) and over 61 (6.1%), in descending order. In terms of marital status, the largest number of the respondents were married (55.7%), followed by unmarried (40.6%), and the lowest number were others (3.7%). It was found that men were more willing to travel abroad, and most of the respondents were 51–60 years old and married.

### 4.2. Analysis of Travel Decisions, Risk Perception, Health Beliefs, Physical and Mental Health, and Travel Intentions

The main purpose of policy development is to solve problems, and the feelings of the public can truly reflect its effectiveness [10,48]. Although tourism can improve physical and mental health [37,40] and personal health beliefs, which can enhance the effectiveness of personal prevention [55], the risk of infection still exists [48] and may still affect travel intentions [6]. Therefore, we analyzed and verified the consistency of people’s perceptions of travel decisions, risk perceptions, health beliefs, physical and mental health, and travel intentions using statistical tests, as shown in Table 3. It was found that, in terms of travel decision making, the greatest number of people agreed that it was a good travel planning decision (3.46) and the lowest number agreed with the government’s policy of promoting travel bubbles (3.00), which was different from Hypothesis 1. As shown in Table 3.

The respondents said:

Respondent 1 said, “The tourism industry has been stagnant for a long time due to the epidemic. The government’s promotion of tourism bubbles at this time should help the industry recover.”

Respondent 3 said, “The government’s promotion of the tourism bubble will help the industry recover, but the price is too high.”

Respondent 2 said, “The promotion at this time is the right time. If the promotion is good, it should be able to activate the economy, but the price is too high to be affordable by the general public.”

Respondent 4 said, “I believe that the policies are well planned, especially in this environment, but exaggerated prices will reduce people’s willingness.”

After analyzing people’s perceptions of the environmental risk of participating in travel bubble activities, it was found that the possibility of infection in airport transit (4.05) was the highest, and the possibility of infection in entering and leaving outdoor places (3.75) was the lowest, which was different from Hypothesis 2.

The respondents said:

Respondent 1 said, “The objects entering and leaving the airport are too complicated, the method of epidemic transmission is currently not very certain, and the risk of infection is high.”

Respondent 2 said, “Chinese people have a strong awareness of epidemic prevention and are afraid of being infected.”

Respondent 3 said, “Under the threat of the epidemic, outing activities have been reduced, and group activities have been reduced.”

Respondent 5 said, “The spread of COVID-19 has not yet been controlled, and there is still a risk of asymptomatic infection. Fear of infection, keep a safe distance when going out.”

Respondent 4 said, “At present, the anti-epidemic policies of various countries have been gradually improved. They can be controlled by measures such as body temperature and disinfection. Individuals are encouraged to wear masks in and out of public places, and to maintain awareness of distance and other awareness.”

In terms of health beliefs, the knowledge of infection prevention behaviors (3.93), the infectiousness and lethality of individuals with positive test results (3.63), and the ability to recognize symptoms of viral infection (3.36) were the highest, and “I would die if diagnosed” (2.70) was the lowest, which was different from Hypothesis 3.

The respondents said:

Respondent 3 said, “After many influenza and virus crises, Taiwanese people have taken a highly subconscious approach to epidemic prevention.”

Respondent 4 said, “The epidemic prevention behavior of the Taiwanese people has become a lifestyle habit.”

Respondent 5 said, “The epidemic has not yet been resolved, and infections have appeared one after another, even the virus has mutated, and fatal cases have spread frequently. So far, it is impossible to accurately grasp the route of infection.”

Respondent 2 said, “Tourists who travel abroad know the severity and risks of the epidemic abroad, and few people will travel abroad at this time.”

Respondent 6 said, “Taiwan’s public health education has been promoted for many years, coupled with the experience of SARS crisis, the people know the seriousness of the epidemic.”

Respondent 1 said, “After many epidemic preventions, the current epidemic control in Taiwan is stable. However, it may also be stable for a long time and begin to relax, leading to actions that violate the code of conduct for epidemic prevention one after another.”

Regarding the physical and mental health components, being enthusiastic (2.62), relieving headaches or stress in the head (2.57), the cessation of anxiety, and temper tantrums (2.45) were the highest, while increasing job performance satisfaction (2.53), relieving back pain (2.39), and the cessation of stomach pain and indigestion (2.35) were the lowest, with results which were dissimilar to Hypothesis 4.

The respondents said:

Respondent 4 said, “Traveling can help relieve stress. I haven’t traveled for a long time. However, the risk of transmission of asymptomatic infected persons in the current epidemic is worrying.”

Respondent 2 said, “At the airport, there are many people or citizens from different countries returning to Taiwan, and they are worried about their health and the risk to the current situation in Taiwan.”

Respondent 3 said, “Traveling can improve physical and mental health. At this time, the epidemic environment, coupled with work pressure, is more looking forward to travel to revitalize the body and mind (A4).”

Respondent 1 said, “If you travel out, you still worry about the epidemic, especially now that the world has not completely suppressed the virus, and the airport has tourists from different countries entering and leaving the airport.”

Respondent 6 said, “The fatality rate of the epidemic has soared, and the risk of asymptomatic infections and other issues will cause pressure on the public. The pressure will affect life behavior and judgment.”

Respondent 5 said, “Sensitive moments can create a sense of oppression. This is invisible, subconscious pressure. Apart from worry, it is impossible to obtain a regular and proper lifestyle.”

Finally, when analyzing people’s willingness to participate in the travel bubble campaign, it was found that the highest intention was to continuously collect relevant information in the future (2.91) while the lowest was to have a high likelihood of participation in the future (2.61), which is different from Hypothesis 5.

The respondents said:

Respondent 3 said, “Tourism can help people improve their physical and mental health, especially at this time of the epidemic environment, coupled with work pressure, the people are looking forward to it.”

Respondent 1 said, “The epidemic cannot be resolved, and there are multiple ways of infection, and there are tourists from other countries entering and leaving the airport.”

Respondent 5 said, “Sensitive moments can create a sense of oppression. This is invisible, subconscious pressure. Apart from worry, it is impossible to obtain regular and proper travel behaviors. This will make the public still maintain a wait-and-see attitude.”

Respondent 2 said, “Many people still remain highly skeptical about the safety of traveling.”

### 4.3. Analysis of the Correlation among Travel Decision Making, Risk Perception, Health Beliefs, Physical and Mental Health, and Travel Intentions

The effectiveness of a policy decision is best demonstrated by the public’s response [10,48]. The ability to accurately predict risk [60], the possession of sound health beliefs and behaviors [56], and obtaining positive physical and mental health perceptions [10] should increase the willingness to participate in a policy [37,54,55,64]. Therefore, Pearson’s test was used to analyze and verify the effects of travel decision making, risk perception, health beliefs, physical and mental health, and travel intentions, as shown in Table 4. The analysis revealed that travel decisions (0.634) were moderately positively correlated with travel intentions (*p* < 0.001); physical and mental health assessment (0.716) was highly positively correlated with travel intentions (*p* < 0.001); and environmental risk (−0.130) was negatively correlated with travel intentions (*p* < 0.05), with consistent results with Hypotheses 6, 7 and 9. Health beliefs, on the other hand, had no significant effect on travel intentions (*p* > 0.05), which is inconsistent with Hypothesis 8. As shown in Table 4.

The respondents said:

Respondent 3 said, “Although I am worried about the risk of infection, I think the tourism decision planned by the government is a proper measure, which can actually improve the industry and people’s livelihood issues, and should be trusted by the public.”

Respondent 1 said, “As the epidemic cannot be resolved and the infection methods are diverse, there are tourists from other countries entering and leaving the airport. If the risk can be reduced, the willingness of passengers to sign up should be improved.”

Respondent 4 said, “Although the infection route of the epidemic is clear, asymptomatic infected persons are also at risk of spreading the virus.”

Respondent 2 said, “The risk of infection is indeed one of the factors to consider participating in activities.”

Respondent 5 said, “Many people still remain highly skeptical about the safety of traveling. No matter how perfect the epidemic prevention is, the unpredictable risk of infection will still affect the willingness to travel.”

Further analysis of the travel decisions revealed significant correlations (*p* < 0.001) between the willingness to improve the overall domestic economy of travel (0.580), the willingness to participate in leisure time (0.584), the high likelihood of future participation (0.533), the continuous collection of relevant information in the future (0.464), and the ability to improve one’s ability to travel (0.541).

In terms of environmental risk, there was a low negative correlation between possible infection at the airport and the willingness to use leisure time to participate (−0.127), a high likelihood of future participation (−0.146), and a significant correlation (*p* < 0.001) between possible infection in accommodation and meals (−0.156) and the willingness to use leisure time to participate.

In terms of physical and mental health, there was a significant correlation between enthusiasm and the willingness to travel (0.741), the willingness to participate in leisure time (0.698), a high likelihood of future participation (0.738), and efforts to improve one’s ability to travel (0.676). There was a significant (*p* < 0.001) relationship between increased work efficiency and the continued collection of relevant information in the future (0.603). As shown in Table 5.

The respondents said:

Respondent 1 said, “The epidemic has affected the tourism industry for a long time, and policy promotion has helped the industry recover. However, the airport is complicated to enter and exit, the method of infection is uncertain, and the risk of infection is high. If the risk is reduced, it should increase the willingness of passengers to sign up.”

Respondent 2 said, “The promotion should be able to activate the economy, but the price is too high to be affordable by the general public.”

Respondent 4 said, “The policies are believed to have been properly planned, especially in this environment, but exaggerated prices will reduce the willingness of the people. At present, epidemic prevention policies in various countries have been improved, and measures such as body temperature and disinfection can be used to control and encourage individuals to enter and leave the public. Wear masks in places and keep a distance, and the Chinese people are more aware of epidemic prevention.”

Respondent 3 said, “Traveling helps people improve their physical and mental health, especially at this time of the epidemic environment, coupled with work pressure, people look forward to traveling to revitalize their physical and mental health. Improving measures can actually improve the problem and make the people trust.”

Respondent 5 said, “Many people still remain highly skeptical about the safety of traveling. No matter how perfect the epidemic prevention is, the unpredictable risk of infection will still affect the willingness to travel.”

### 4.4. Validation of the Regression Analysis of the Effects of Travel Decisions, Environmental Risk Perception, Health Beliefs, and Physical and Mental Health Perceptions on Travel Intention

Based on the above analysis, it was found that travel decisions, environmental risks, and physical and mental health perceptions were significantly correlated with travel intentions. Then, regression analyses were conducted to identify the fitness of the model for the effects of travel decisions, environmental risks, and physical and mental health perceptions on travel intentions. The analysis showed a df value of 4, an F value of 98.392, and an explanatory power of 61.6% (corrected R2). Travel decisions (0.361) and physical and mental health perceptions (0.533) were significantly and positively associated with travel intentions (*p* < 0.01). There was no significant effect of environmental risk perceptions and health beliefs (*p* > 0.01). As shown in Figure 2.

Among them, the issues of improving the overall domestic economy (0.247), physical and mental health awareness to increase motivation (0.379), and the reduction of headaches and stress (0.251) had a significant positive effect on travel decisions (*p* < 0.01), with an explanatory power of 65.6% (modified R2).

The respondents said:

Respondent 1 said, “The epidemic has affected the tourism industry for a long time, and policy promotion has helped the industry recover. However, the airport is complicated to enter and exit, the method of infection is uncertain, and the risk of infection is high. If the risk is reduced, it should increase the willingness of passengers to sign up.”

Respondent 3 said, “Traveling helps people improve their physical and mental health, especially at this time of the epidemic environment, coupled with work pressure, people look forward to traveling to revitalize their physical and mental health. Improving measures can actually improve the problem and make the people trust.”

Respondent 4 said, “The policies are believed to be properly planned, especially in this environment.”

Respondent 5 said, “Many people still remain highly skeptical about the safety of traveling. No matter how perfect the epidemic prevention is, the unpredictable risk of infection will still affect the willingness to travel.”

### 4.5. Discussion

#### 4.5.1. Research Samples and Limitations

We believe that although women like to travel, they are cautious and avoid the risks of travel. People over the age of 51 are more likely to have families and jobs, and to have worked for many years, accumulating enough wealth to provide the time and money to travel abroad in order to broaden their horizons and improve their quality of life. Therefore, men are more willing to travel abroad, especially those who are aged 51–60 years old and married.

#### 4.5.2. Travel Decisions, Risk Perceptions, Health Beliefs, Physical and Mental Health, and Travel Intentions

The study concluded that although policy-making can help government agencies solve development challenges [10,48], policy formulation is flawed and price setting is not in line with the market demand, resulting in the low public recognition of the policy. As a result, the public perceives that travel policy planning is better, while policy promotion is not effective.

Although countries are gradually improving their epidemic prevention measures by implementing such measures as body temperature screening and alcohol disinfection in public places or hotels, the risk and the lethality of virus infection is high [17,18], and the airport is full of miscellaneous people entering and leaving the country. Therefore, the public perceives a high risk of infection in airports but a low risk when entering and leaving outdoor public places.

The epidemic is still unresolved, and there is still a risk of infection and even death among those with asymptomatic infection, but increased searching for health information and attention to personal health [55] can reduce the risk of accidents among individuals [57,58]. In addition, the national epidemic prevention measures and experience are sufficient, and the individuals have sufficient experience in epidemic response measures. As a result, people are aware of the virus, are not afraid of the high transmission and mortality rates or the threat of death from confirmed diagnosis, and understand how to prevent the disease, but are unable to recognize the symptoms of the infection with certainty.

Although travel is expected to stimulate the passion for life, relieve stress, and relax the body and mind [28], the global epidemic has not yet subsided, the airport is full of people with complex backgrounds, and there are multiple routes of virus transmission and high mortality rates, so it is still risky to engage in travel behaviors at this time [6]. Therefore, people believe that participating in travel may increase their passion, relieve headaches and stress, and relieve their emotions, but it still does not improve back pain, resolve indigestion, improve work efficiency, or increase satisfaction.

In an epidemic environment and under work pressure, people look to tourism to improve their physical and mental health [28]. Moreover, promoting people’s participation in tourism activities can promote industrial development [36]. However, the complex background of people entering and leaving the airport, the unresolved epidemic, and the multiple modes of transmission create an invisible sense of oppression at sensitive times, leading to a wait-and-see attitude toward participation in tourism activities. As a result, people indicated that they would continue to collect information but would not be very willing to participate.

#### 4.5.3. Correlation between Travel Decisions, Risk Perception, Health Beliefs, Physical and Mental Health, and Travel Intentions

People who highly agree with the policy, and have the ability to assess risks, maintain high health beliefs, and maintain positive feelings of physical and mental health should have an increased willingness to participate in the policy implementation [10,37,48,54,55,60,64]. However, unresolvable epidemic problems, high risk, and the high price of activities promoted by travel policies are all factors that affect travel intentions. This resulted in significant correlations between travel decisions, physical and mental health assessments, environmental risks, and travel intentions.

Furthermore, countries are gradually improving their epidemic prevention mechanisms, and the monitoring and epidemic prevention measures in public places such as hotels and restaurants are becoming more and more stringent. People also expect tourism to promote their physical and mental health and revitalize the economy. However, factors such as long airport routes, diverse tourist backgrounds, the mixed backgrounds of hotel and restaurant diners, as well as multiple modes of virus transmission, the high risk of infection, and high death rates still affect people’s willingness to travel abroad and lead to a negative impact on their physical and mental health. This has resulted in a wait-and-see attitude of the people toward short-haul cross-country travel policies and an inability to effectively improve their physical and mental health. Therefore, issues such as the overall improvement of the economy, the possibility of infection at airports, the possibility of infection in accommodation and restaurants, passion, and increased work efficiency have a relevant impact on the desire to travel.

#### 4.5.4. Regression Analysis of Travel Decisions, Environmental Risk Perceptions, Health Beliefs, and Physical and Mental health Perceptions on Travel Intentions

The investigators concluded that travel activities can relieve stress [28] and enable people to achieve some degree of health improvement. The travel bubble decision is a short-haul, country-to-country overseas travel activity planned by the Taiwanese government in cooperation with low-risk countries, and tourists must comply with the epidemic prevention measures for departure and return. Moreover, Taiwanese people are highly aware of epidemic prevention, and the epidemic in Taiwan remains under control, which makes people trust the “Travel Bubble” decision.

However, the current epidemic has multiple routes of infection, high mortality [17,18], and increased uncertainties such as variant viruses and asymptomatic patients [17,18,25]. The public’s willingness to plan outdoor activities or to travel abroad has varied. Nevertheless, due to the prolonged depression generated by the epidemic [43,44], safe and reliable tourism opportunities certified and planned by the government [28] are expected to alleviate the depression of the epidemic, promote tourism, and revitalize the domestic economy through tourism.

In contrast to general travel plans and needs, the uncertainty of the infection risk and the differences in levels of expertise led to the belief in the government decisions and the effectiveness of the epidemic prevention, hands-on travel experiences, and the expectation of rapid improvements in physical and mental health as the main key factors influencing travel intentions. As a result, environmental risk perceptions, health beliefs, and travel intentions do not have a significant impact. However, travel decision-making and physical and mental health perceptions have a significant impact on travel intentions, with factors such as increasing enthusiasm, reducing headaches and stress, and improving the overall domestic economy having the greatest impact.

## 5. Conclusions

The study found that travel bubble events were successful because participants were aware of the seriousness of the infection, understood how to prevent it, did not fear the threat of death if diagnosed, and looked to travel to boost their personal enthusiasm, relieve headaches and stress, and soothe emotions. However, the transmission rates and mortality rates are high, the transmission modes are diverse, and there is no way to clearly identify asymptomatic patients. The background of people entering and leaving airports, hotels, and restaurants is complex. Participation in travel activities may create a sense of oppression. The policy is flawed, and the prices are not in line with the market demand. These factors reduce public acceptance of the travel bubble policy, leading to a wait-and-see attitude about participating in the next travel bubble event. Factors such as increasing enthusiasm, reducing headaches and stress, and improving the overall domestic economy will be key predictors of people’s decisions to participate in travel bubbles.

Based on the above, the following suggestions are offered:For Governments and PolicymakersIn addition to incorporating epidemic prevention measures, travel policies can utilize cell phone applications for locational control. Vaccinations and screenings must be designed to meet the needs of consumers. Without proper planning, decisions will stagnate and administrative resources will be wasted. Therefore, we suggest that the government should establish a consensus with the tourism industry and design a price level that is in line with the market mechanism and the cost in order to increase the willingness to participate.For Travel AgenciesIt is a basic consideration for travel agencies to plan their itineraries based on cost, but it is important not to discourage consumers by killing the goose that lays the golden egg. Therefore, it is suggested that travel itineraries should be planned with cost in mind, and that prices should be adjusted or travel itineraries should be suspended in a timely manner in order not to affect the image of the company by causing poor perceptions.Suggestions for Future ResearchThe study was conducted with Taiwanese people as the target population in order to explore the willingness to participate in the promotion of the travel bubble policy. Due to the limitations of the study, there are shortcomings in the research method and sample size. It is suggested to expand the sample size of the questionnaire and use additional validation software or computational methods to investigate the correlation between travel decisions, risk perception, physical and mental health, and other issues that can fill the research gap.

## Figures and Tables

**Figure 1 ijerph-18-05717-f001:**
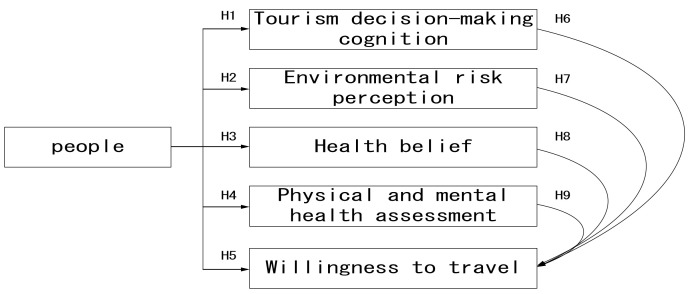
The research structure.

**Figure 2 ijerph-18-05717-f002:**
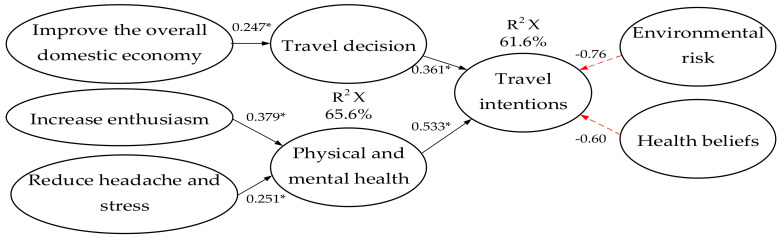
Results of the regression analysis of travel decisions, environmental risk perceptions, health beliefs, and physical and mental health perceptions on travel intentions. * *p* < 0.05.

**Table 1 ijerph-18-05717-t001:** Analysis of the questionnaire tools.

Construct	Dimension	
Basic Variables	Experience abroad (Yes (continue to answer), no (end of answer)), gender (male: female), age (20 down: 21–30:31–40:41–50:51–60:61 up), Marital status (single, married, other)	
Construct	Dimension	Cronbach’s α
Tourism decision-making cognition (0.921)	Agree with the government’s promotion of tourism bubble policy	0.915
	Agree with relevant government agencies’ plans for epidemic prevention and tourism	0.903
	Agree with travel agency related travel itinerary arrangements and planning	0.910
	Acknowledge relevant government subsidies	0.908
	Recognition is a proper tourism planning decision	0.904
	Acknowledgement can improve the overall domestic economic dilemma	0.914
	Recognition can improve the plight of the tourism industry	0.911
Environmental risk perception (0.922)	May be infected with the new coronavirus during airport transfers	0.919
	You may be infected with the new coronavirus when taking public transportation	0.898
	You may be infected with the new coronavirus when entering and leaving indoor tourist places	0.897
	Entering and exiting outdoor tourist places may be infected with the new coronavirus	0.915
	You may be infected with the new coronavirus when you are dining in the hotel	0.902
	Outgoing shopping and consumption may be infected with the new coronavirus	0.913
Health belief (0.760)	If diagnosed with the virus, I will die	0.710
	Infected with the virus, will die	0.699
	Physical health, high risk of death after infection	0.708
	Shows a positive response to the virus, which can be transmitted to relatives and friends and cause death	0.707
	Participate in activities, there is a high risk of infection	0.720
	Participate in activities, the chance of exposure to the virus is high	0.737
	I can take measures to prevent infection	0.725
	I know how to prevent infection	0.727
	If I come into contact with the virus, I know what to do	0.743
	I am sure to understand the health guidelines for virus prevention	0.724
	Can identify the symptoms of the virus	0.754
Physical and mental health awareness (0.980)	Can increase my satisfaction with work performance	0.974
	Can make me passionate about things or activities	0.974
	Can make me feel that I increase my work efficiency	0.974
	It relieves me of headaches or pressure on top of my head	0.974
	Can relieve me of backache	0.972
	So I don’t have insomnia or a good night’s sleep	0.972
	So that I don’t feel stomach pain and indigestion	0.973
	Can restore my appetite	0.973
	Can make me no longer anxious and lose my temper	0.973
Willingness to participate (0.940)	Willing to participate in free time	0.911
	The willingness to participate in the future is very likely	0.904
	Continue to collect relevant information in the future	0.935
	Make every effort to improve the abilities required for personal travel	0.909

**Table 2 ijerph-18-05717-t002:** Background analysis and topics of the interview.

A1	A2	A3	A4	A5	A6
Guide	Leader	Leisure Prof	Tourism Prof	Dr.	Public health Dr.
What is your opinion on the current state of tourism bubble policy planning? What is your willingness to participate? What are the main reasons for the analysis results? If you participate in the formation of the tourism bubble, what are the risks you are worried about? What are the main reasons for the analysis results? What do you think are the most strengthened health beliefs and physical and mental health indicators for individuals facing the epidemic environment? Those are the weakest? What are the main reasons for the analysis results?					

**Table 3 ijerph-18-05717-t003:** Analysis of travel decisions, risk perception, health beliefs, physical and mental health, and travel intentions. Analysis of intentions.

Issue			M	Rank
Travel decisions	Agree with the government’s promotion of tourism bubble policy		3.00	7
	Agree with government agencies’ epidemic prevention and tourism planning		3.19	5
	Agree with travel agency itinerary and planning		3.20	4
	Acknowledge relevant government subsidies		3.38	2
	Recognition is a proper tourism planning decision		3.46	1
	Acknowledgement can improve the overall domestic economic dilemma		3.11	6
	Recognition can improve the plight of the tourism industry		3.27	3
Risk perception	Airport transfer may be infected		4.05	1
	You may be infected when taking public transportation		3.99	2
	May be infected when entering and leaving indoor places		3.93	3
	May be infected when entering and leaving outdoor places		3.75	6
	May be infected while eating		3.87	4
	Outgoing shopping consumption may be infected		3.84	5
Health beliefs	Perceived susceptibility and seriousness	Diagnosed, I will die	2.70	6
		Infection, death	3.07	4
		High risk of death after infection	2.95	2
		Showing positive reaction, contagious and lethal	3.63	1
		High risk of infection	3.36	3
		High chance of participating in exposure to the virus	3.50	2
	Self-efficacy	Can take measures to prevent infection	3.86	2
		Know to prevent infection	3.93	1
		Contact the virus and know what to do	3.86	2
		Understanding the prevention guidelines	3.77	3
		Can distinguish virus symptoms	3.36	4
Physical and mental health	Psychology	Increase job performance satisfaction	2.53	3
		Passionate	2.62	1
		Increase work efficiency	2.59	2
	Mental status	Relieve headaches or overhead pressure	2.57	1
		Reduce backache problem	2.39	3
		Improve sleep quality	2.41	2
	Life attitudes and health	No stomachache and indigestion	2.35	3
		Restore appetite	2.40	2
		No longer anxious, lose your temper	2.45	1
Travel intentions	Willing to participate in free time		2.66	3
	The willingness to participate in the future is very likely		2.61	4
	Continue to collect relevant information in the future		2.91	1
	Make every effort to improve the abilities required for personal travel		2.76	2

**Table 4 ijerph-18-05717-t004:** Correlation analysis of travel decisions, risk perception, health beliefs, physical and mental health, and travel intentions.

Facet	Travel Decision	Environmental Risk	Health Beliefs	Physical and Mental Health
Travel intentions	0.634 **	−0.130 *	−0.103	0.716 **

* *p* < 0.05; ** *p* < 0.01.

**Table 5 ijerph-18-05717-t005:** Analysis of the correlation between travel decisions, risk perception, health beliefs, physical and mental health, and travel intentions.

Issue	Total Dimensions of Travel Willingness	Is Willing to Participate in Free Time	Is Likely to Participate in the Future	Will Continue to Collect Relevant Information in the Future	Will Try to Improve the Ability Required for Personal Travel
Travel decisions	0.634 **				
Agreeing with government policies	0.502 **	0.491 **	0.476 **	0.388 **	0.484 **
Agreeing with government planning	0.537 **	0.528 **	0.515 **	0.420 **	0.504 **
Agreeing to travel agency arrangements	0.512 **	0.514 **	0.519 **	0.313 **	0.529 **
Supporting government subsidy measures	0.510 **	0.486 **	0.491 **	0.352 **	0.537 **
Proper travel policies	0.494 **	0.481 **	0.469 **	0.375 **	0.483 **
Will improve the overall domestic economy	0.580 **	0.584 **	0.533 **	0.464 **	0.541 **
Will solve the difficulties of the tourism industry	0.528 **	0.545 **	0.495 **	0.393 **	0.504 **
Environmental risks	−0.130 *				
Possible infection at the airport	−0.127 *	−0.150 *	−0.146 *	−0.054	−0.117
Possible infection in public transport	−0.108	−0.125	−0.139 *	−0.034	−0.101
Possible infection at indoor travel venues	−0.077	−0.091	−0.108	−0.009	−0.077
Possible infection at outdoor travel venues	−0.038	−0.094	−0.062	0.034	−0.02
Possible infection at accommodation and restaurants	−0.1	−0.156 *	−0.135 *	0.013	−0.091
Possible infection at shopping trips	0.007	−0.036	−0.018	0.049	0.029
Physical and mental health	0.716 **				
Increased job satisfaction	0.660 **	0.632 **	0.663 **	0.519 **	0.604 **
Passionate	0.741 **	0.698 **	0.738 **	0.600 **	0.676 **
Increased work efficiency	0.717 **	0.657 **	0.696 **	0.603 **	0.670 **
Reduced headaches or stress	0.716 **	0.673 **	0.698 **	0.578 **	0.671 **
Relieving back pain	0.611 **	0.594 **	0.600 **	0.483 **	0.561 **
Improved sleep quality	0.626 **	0.596 **	0.617 **	0.495 **	0.584 **
No more stomach pains or indigestion	0.575 **	0.536 **	0.569 **	0.446 **	0.555 **
Restored appetite	0.611 **	0.590 **	0.579 **	0.499 **	0.570 **
No longer anxious and angry	0.645 **	0.637 **	0.607 **	0.524 **	0.592 **

* *p* < 0.05, ** *p*< 0.01.

## Data Availability

No data support.

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
