# Peer review of "Is the Travel Bubble under COVID-19 a Feasible Idea or Not?"

_ijerph, 2021, doi:10.3390/ijerph18115717_

Round 1
Reviewer 1 Report
Dear Authors,
The paper presents the results of a very interesting study and it is very timely and of the potential impact on policymakers and the tourism industry. In general, it is well written and structured. However, there are some points that are not adequately clearly presented and should be considered for better clarifications.
- There is a need to define (maybe as a footnote) what is a "travel bubble", for some readers it is a new concept.
- There is a reference to Palau as a chosen destination (by the Taiwan government). It is interesting if it is the only destination? If so why? Was there any intergovernmental deal on it? Moreover, I would advise (again maybe in a footnote) to explain what Palau is - for most European or North American readers this name does not mean anything, they do not know where it is. Such knowledge is important to assess its meaning for the travel bubble concept.
- In table 1 there is information that the basic variable was experience abroad (no experience end of answer). It should be also provided when there is a sample characteristic - that it does not regard the general population but only people who are travelling abroad? It is not clear if it regards general travelling (I assume yes) or just now in the pandemic situation?
- There is a very confusing statement at page 7 "The study was conducted to investigate the water literacy of people in Taiwan" - I do not understand - is it by mistake? Or was this study part of another project? It needs to be cleared up.
- The table 3 needs to be rebuild or reformatted. In its current shape issues of mental health or travel intentions looks like a part of travel decision issues or risk perception (they are below and not bolded). There is a need to provide description what is here symbol M and what are the ranks. Not all readers will understand this table without them..
- When the results are discussed there is a need for reflection on the timing of survey. We know that willingness to travel and to take a holiday change across the year and the results are also dependent on the covid situation or media coverage on it when the survey was conducted.
- For an international usage or comparison, it also would be valuable to make some references in the literature review on general health attitudes of Taiwan inhabitants as compared to other nations. Are they more conscious? More precautions?
Best regards
Author Response
Reviewer 1
- There is a need to define (maybe as a footnote) what is a "travel bubble", for some readers it is a new concept.
We will add it in the text. Such as line 48-57.
- There is a reference to Palau as a chosen destination (by the Taiwan government). It is interesting if it is the only destination? If so why? Was there any intergovernmental deal on it? Moreover, I would advise (again maybe in a footnote) to explain what Palau is - for most European or North American readers this name does not mean anything, they do not know where it is. Such knowledge is important to assess its meaning for the travel bubble concept.
We will add it in the text. Such as line 48-57.
In table 1 there is information that the basic variable was experience abroad (no experience end of answer). It should be also provided when there is a sample characteristic - that it does not regard the general population but only people who are travelling abroad? It is not clear if it regards general travelling (I assume yes) or just now in the pandemic situation?
The characteristics of the sample mainly explain the basic information of the subjects in the manuscript. The relevant instructions are as shown in lines 212-222.
- There is a very confusing statement at page 7 "The study was conducted to investigate the water literacy of people in Taiwan" - I do not understand - is it by mistake? Or was this study part of another project? It needs to be cleared up.
We have deleted this paragraph.
- The table 3 needs to be rebuild or reformatted. In its current shape issues of mental health or travel intentions looks like a part of travel decision issues or risk perception (they are below and not bolded). There is a need to provide description what is here symbol M and what are the ranks. Not all readers will understand this table without them.
We have added explanations in the manuscript. Such as lines 311-315.
- When the results are discussed there is a need for reflection on the timing of survey. We know that willingness to travel and to take a holiday change across the year and the results are also dependent on the covid situation or media coverage on it when the survey was conducted.
We have added explanations in the manuscript. Such as line 560-575.
- For an international usage or comparison, it also would be valuable to make some references in the literature review on general health attitudes of Taiwan inhabitants as compared to other nations. Are they more conscious? More precautions?
We have added explanations in the manuscript. Such as lines 112-113 and lines 560-575.
Thank you reviewer
Your suggestions will make the manuscript more complete.
Reviewer 2 Report
Dear authors, the paper is interesting. However, I suggest major revisions. Please take into consideration the comments below.
1. The title is too long. I suggest simplifying: “Is the travel bubble under COVID-19 a feasible idea or not?”
2. In the introduction, the authors say that “Due to the experience in epidemic prevention, there is no local outbreak in Taiwan and the situation is well controlled [13].”
However, later the authors say “The government proposed a travel bubble policy in March 2021 to select regions with good epidemic control and to target short-distance country-to-country travel activities with Palau as the destination [18].” Please clarify why Palau was excluded as the destination. Was there any restriction to travel to China?
3. In section 3.2, KMO should be > than 0.5 instead of 0.06?
4. The authors say that:
“There were 7 questions on travel decision cognition…”
“There were 6 questions on environmental risk perception…”
“There were 11 questions on health beliefs…”
“There were 9 questions on physical and mental health assessment…”
“There were 4 questions on travel intention…”
The questions are listed in Table 1, however, the data is not presented. I suggest the authors add an Excel file with the 560 answers, already with scores 1 to 5. Otherwise, it is not possible to verify the results.
5. In Section 3.3, the authors say: ” Due to the research approach and limitations, the study sample size and results may be inadequate.” Please remove the phrase, otherwise, the paper results cannot be trusted, and therefore, the paper cannot be published.
6. I understand there were limitations to collect the questionnaires, and the authors present those very well. Still, please answer these questions:
6.1 Why only 600 questionnaires were collected (560 validated) Is this number representative of Taiwan population?
6.2 Was the online survey conducted at a national level, or mostly to a community/city?
6.3 Please indicate the dates that the questionnaires were collected.
7. In Table 2 there are presented several experts. How many experts did the authors have interviewed?
8. Section 4.2 is quite confusing to me. Maybe the authors can group all the answers provided by each expert (Guides (A1), Leaders (A2), Leisure Prof (A3), Tourism Prof (A4), Dr (A4), and Public health dr (A5)) in a more organized way.
9. Please clarify the way the sentence is presented. Do experts A3 have the same opinion as A1? “A1: The tourism industry has been stagnant for a long time due to the epidemic, and the government's promotion of travel bubbles at this time should help the industry recover (A3).”
10. Tables 3 and 5 are difficult to follow. I suggest the authors compile all data and results on an Excel file. Moreover, is confusing to present A1, A2…, A5 with different meanings in Table 2 and Table 5.
11. SPSS is a good software and the authors could explore more the outputs available on the software. I would like to see more results for the Descriptive Statistics, Linear and Multi-linear regression (and discuss if the values of R2 are acceptable). I also would like to see more results of the factorial analysis such as eigenvalues and scores. It would be very interesting if the authors would also apply cluster analysis and include the Dendrogram map.
12. In section 1 of the Conclusion, please explain what do you mean by “sound measures”;
13. Please rephrase “..sacrificing the goose that lays the eggs”.
14. The conclusions are too general and do not have any comment about COVID. The question of the title (Is the travel bubble under COVID-19 a feasible idea or not?) seems it was lost in the paper. Please reformulate the interpretation of the results and conclusions.
15. The 9 hypotheses presented in Figure 1 also were lost in the paper. Please reformulate.
Author Response
Reviewer 2
Dear authors, the paper is interesting. However, I suggest major revisions. Please take into consideration the comments below.
- The title is too long. I suggest simplifying: “Is the travel bubble under COVID-19 a feasible idea or not?”
Already edited. Such as line 1.
- In the introduction, the authors say that “Due to the experience in epidemic prevention, there is no local outbreak in Taiwan and the situation is well controlled [13].”
However, later the authors say “The government proposed a travel bubble policy in March 2021 to select regions with good epidemic control and to target short-distance country-to-country travel activities with Palau as the destination [18].” Please clarify why Palau was excluded as the destination. Was there any restriction to travel to China?
Explained, as in line 44.
- In section 3.2, KMO should be > than 0.5 instead of 0.06?
Amended, as in line 235.
- The authors say that:
“There were 7 questions on travel decision cognition…”
“There were 6 questions on environmental risk perception…”
“There were 11 questions on health beliefs…”
“There were 9 questions on physical and mental health assessment…”
“There were 4 questions on travel intention…”
The questions are listed in Table 1, however, the data is not presented. I suggest the authors add an Excel file with the 560 answers, already with scores 1 to 5. Otherwise, it is not possible to verify the results.
The relevant data has been described in Table 1.
- In Section 3.3, the authors say: ” Due to the research approach and limitations, the study sample size and results may be inadequate.” Please remove the phrase, otherwise, the paper results cannot be trusted, and therefore, the paper cannot be published.
Deleted, such as lines 302-303.
- I understand there were limitations to collect the questionnaires, and the authors present those very well. Still, please answer these questions:
6.1 Why only 600 questionnaires were collected (560 validated) Is this number representative of Taiwan population?
6.2 Was the online survey conducted at a national level, or mostly to a community/city?
6.3 Please indicate the dates that the questionnaires were collected.
The case of this study is the case of Taiwanese government authorities' expected overseas tourism activities led by the government during April 1, 2021. Due to the fact that the number of places open for the event is from all over Taiwan, it is limited by the actual number of participants and personal qualifications, so it is expanded to people who are willing to participate. In the end, only 600 people who meet the qualifications of the test can help to fill out the questionnaire. And use the number of questionnaires as the basis for research inferences. Such as line 218-222.
- In Table 2 there are presented several experts. How many experts did the authors have interviewed?
The study interviewed 6 scholars with professional experience of tour guides and team leaders, as well as leisure, tourism, sightseeing and medical backgrounds as the subject of expert interviews. As shown in Table 2.
- Section 4.2 is quite confusing to me. Maybe the authors can group all the answers provided by each expert (Guides (A1), Leaders (A2), Leisure Prof (A3), Tourism Prof (A4), Dr (A4), and Public health dr (A5)) in a more organized way.
We have re-introduced the presentation of the interview content.
- Please clarify the way the sentence is presented. Do experts A3 have the same opinion as A1? “A1: The tourism industry has been stagnant for a long time due to the epidemic, and the government's promotion of travel bubbles at this time should help the industry recover (A3).”
This statement is representative, and both interviewees said that "the promotion of the travel bubble should contribute to the recovery of the tourism industry."
- Tables 3 and 5 are difficult to follow. I suggest the authors compile all data and results on an Excel file. Moreover, is confusing to present A1, A2…, A5 with different meanings in Table 2 and Table 5.
We have rewritten the contents of Table 2 and Table 5.
- SPSS is a good software and the authors could explore more the outputs available on the software. I would like to see more results for the Descriptive Statistics, Linear and Multi-linear regression (and discuss if the values of R2 are acceptable). I also would like to see more results of the factorial analysis such as eigenvalues and scores. It would be very interesting if the authors would also apply cluster analysis and include the Dendrogram map.
We add regression analysis to explore the impact of tourism decision-making and environmental risk perception, health beliefs, and physical and mental health perceptions on travel intentions. See Figure 2 and lines 468-476.
- In section 1 of the Conclusion, please explain what do you mean by “sound measures”;
We have rewritten the presentation. Such as lines 557-558.
- Please rephrase “..sacrificing the goose that lays the eggs”.
We revised it to "kill the goose that lays the golden egg"
- The conclusions are too general and do not have any comment about COVID. The question of the title (Is the travel bubble under COVID-19 a feasible idea or not?) seems it was lost in the paper. Please reformulate the interpretation of the results and conclusions.
We have rewritten the content of the conclusion. As in line 490-496.
- The 9 hypotheses presented in Figure 1 also were lost in the paper. Please reformulate.
We have the results of the analysis stating Hypothesis 9 in lines 415-416. And discussed in 526-545.
Thank you reviewer
We have completed the correction, based on your suggestions. Your suggestions will make the manuscript more complete.
Reviewer 3 Report
The article deals with an important issue in the current context from the point of view of tourism, travel preference, and personal perceptions, with statistical analysis. The authors assume that the study has many limitations in relation to the sample's representativeness, but at least they should present the sampling error and level of significance reached with the 560 questionnaires answered. Likewise, they should indicate the stratification of the sample carried out, and whether it is consistent with the distribution of the population surveyed according to the gender and age, at least, as well as in terms of geographic distribution on the island of Taiwan.
Moreover, as recommendations for improvement, I also suggest the following points:
- in the Introduction section, authors state that “the transmission route is unknown”, in relation to the COVID-19 disease, but there are many studies that already indicate the transmission routes of the virus through droplets of infected saliva, through contact with contaminated surfaces or, even, through the air.
- the analysis and disccusion section is presented in a very confusing way, mixing the results from the topics of the interview and the hypothesis for the questionnaire answers.
Author Response
Reviewer 3
The article deals with an important issue in the current context from the point of view of tourism, travel preference, and personal perceptions, with statistical analysis. The authors assume that the study has many limitations in relation to the sample's representativeness, but at least they should present the sampling error and level of significance reached with the 560 questionnaires answered. Likewise, they should indicate the stratification of the sample carried out, and whether it is consistent with the distribution of the population surveyed according to the gender and age, at least, as well as in terms of geographic distribution on the island of Taiwan.
We have added explanations and added literature. Such as lines 218-224.
Moreover, as recommendations for improvement, I also suggest the following points:
- in the Introduction section, authors state that “the transmission route is unknown”, in relation to the COVID-19 disease, but there are many studies that already indicate the transmission routes of the virus through droplets of infected saliva, through contact with contaminated surfaces or, even, through the air.
We have rewritten the content of the conclusion and added literature. Such as lines 77-79.
- the analysis and disccusion section is presented in a very confusing way, mixing the results from the topics of the interview and the hypothesis for the questionnaire answers.
We use software to analyze the results of the questionnaire, interview and aggregate the opinions of the interviewees based on the analysis results, and use different formatting statements to interpret the meaning of the questionnaire data.
Thank you reviewer
We have completed the correction, based on your suggestions. Your suggestions will make the manuscript more complete.
Round 2
Reviewer 2 Report
Dear authors,
The paper has improved, the reviewers' suggestions were followed, and the English language and style are fine.
The paper is ready to be published.
Author Response
Dear reviewer
Thank you for your suggestion.
We believe that the manuscript will be more complete.
Reviewer 3 Report
Most of the suggested changes by this referee were implemented adequately in the new version of the paper. Therefore, it can be accepted in the present form.
Author Response

(The authors gave the same response as above.)
